# CMTA: A framework for Multilingual COVID-19 Tweet Analysis

**Raj Ratn Pranesh**[*]
Birla Institute of
Technology
Mesra, India
raj.ratn18@
gmail.com

**Ambesh Shekhar**[*]
Birla Institute of
Technology
Mesra, India
ambesh.sinha@
gmail.com

**Mehrdad Farokhnejad**
Univ. Grenoble Alpes,
CNRS, LIG
Grenoble, France
Mehrdad.Farokhnejad@
univ-grenoble-alpes.fr

## Abstract

In the current scenario of COVID-19 pandemic when people around the world are facing restrictions in their daily life activities, there is a rapid growth in the number of people relying on the internet. Billions of people are using social media platforms such as Twitter for sharing COVID-19 related news, information and thoughts which reflect their perception and opinion about the pandemic. Analysis of tweets for identifying misinformation and emotion analysis can generate valuable insights. We present CMTA: a multilingual COVID-19 related tweet analysis model.In our work, we propose a deep learning model for multilingual tweet misinformation and emotion detection and classification. CMTS uses multilingual BERT for extracting features from multilingual textual data, which is then categorised into specific emotion and misinformation class. Classification is done by a Dense-CNN model trained on tweets manually annotated into emotion(8 classes) and misinformation(5 classes). We also present an analysis of multilingual tweets from April to June month showing the distribution of public emotion and misinformation spread across different languages.

## 1 Introduction

The year 2020 marked the beginning of global catastrophe COVID-19 all around the world. Started from Wuhan, China, it rapidly spread over 214 countries causing hundreds of thousands of cases and effecting human lives to a great extend COVID-19 created a massive impact on multiple sectors which include countries economy, government bodies, private companies, media houses and most importantly effecting the mental and physical health of human beings by tempering their daily routine activities.

COVID-19 also made us realize how well the world is interconnected through the internet and through internet people can perform various task using electronic devices without stepping outside their houses. The increase of people dependency over the internet has resulted in an unparalleled amount of information flow over the internet, most largely through various social media platforms such as Twitter, Facebook, Whatsapp, Instagram, Reddit where people shared their response, thoughts, news, information related to COVID-19. Studies have shown that many people connect to internet and social media platforms everyday and utilizing it for getting information/news through them(Matsa and Shearer, 2018)(Hitlin and Olmstead, 2018).

With such a huge amount of human-generated information being exchanged everyday, it has attracted NLP researchers to explore, analyze, and generate valuable insights about people response to COVID-19 through emotion analysis as well as to identify and remove malicious information through misinformation detection.

In our paper, we present CMTA: a multilingual tweet emotion analysis and misinformation detection model for understanding both the negative and positive sides of social medial during COVID-19 pandemic. CMTA uses Multilingual BERT, trained on 104 multiple languages to derive features from tweets and 1D convolution for finding the correlation between data of hidden states. It also uses a dense layer for linear transformation on contextual embeddings to provide inferential points. Our work helps in providing better results in the classification of tweet's emotion and finding the proximity of being fake. We used manually annotated tweet emotion data and misinformation data for training two separate deep neural network model- (i) training a classifier model for classifying tweets into emotion class and (ii) training a classifier model for detecting and identifying the type of disinformation

---

*equal contribution

present in tweets.

The **motivation** behind designing a multilingual model lies behind the need of analyzing not just monolingual tweets but multilingual tweets by building a single deep learning framework that would be able to understand tweets in multiple languages.

We used our trained models for a systematic analysis of COVID-19 related tweets collected from April 2020 to June 2020. The analysis tweets are done based on the distribution of type of (i) emotion behind the tweets and, (ii) type of misinformation present in tweets with respect to- (i) the language used for writing a tweet, and (ii) the country/region of a tweet's origin. We provided a language-wise survey where for each language, we have investigated the perception of public on Twitter towards COVID-19 and it is related policies by classifying the tweet in eight classes of emotion namely, 'anger', 'disgust', 'fear', anxiety, sadness, happiness, relaxation ,and desire. Along with this, we also investigated the presence of false information spread throughout the Tweeter by classifying the tweets in five classes namely, 'false', 'partly false', 'misleading','true' and 'no evidence'. We have provided illustrative statistical representation of our findings and detailed discussion about the insights discovered in our survey.

In the following sections, we will discuss the data preparation, proposed model, multilingual tweet emotion analysis, multilingual tweet misinformation analysis and conclusion.

## 2   Related Work

This section summarises research done in social media analysis focusing on shared information through Twitter regarding the COVID-19 pandemic.

In the last months, the COVID-19 pandemic has resulted in immense growth in studies that have been published to investigate the impact of the COVID-19 pandemic on Twitter. (Huang and Carley, 2020) examined the global spread of information related to crucial disinformation stories and "fake news" URLs during the early stages of the global pandemic on Twitter. Their study shows that news agencies, government officials, and individual news reporters do send messages that spread widely, and so play critical roles. However, the most influential tweets are those posted by regular users, some of whom are bots. Tweets mentioning "fake news" URLs and misinformation stories are more likely to be spread by regular users than the news or government accounts.

(Sharma et al., 2020) focused on emotion analysis and topic modeling and designed a dashboard to track misinformation on Twitter regarding the COVID-19 pandemic .The dashboard provides an analysis of topics, emotions, and trends, assessed from Twitter posts; along with identified false, misleading and clickbait information spreading on social media, related to COVID-19.

(Singh et al., 2020) are monitoring the flow of (mis)information flow across 2.7M tweets, and correlating it with infection rates to find that misinformation and myths are discussed, but at lower volume than other conversations.They observed that a meaningful spatio-temporal relationship exists between information flow and new cases of COVID-19, and while discussions about myths and links to poor quality information exist, their presence is less dominant than other crisis specific themes.

In (Gencoglu and Gruber, 2020) proposed a first example of causal inference approach to discover and quantify causal relationships between pandemic characteristics (e.g. number of infections and deaths) and Twitter activity as well as public emotion. They observed that their proposed method could successfully capture the epidemiological domain knowledge and identify variables that affect public attention and perception.

An infodemic observatory analysing digital responses in online social media to COVID-19 has been created by CoMuNe lab at Fondazione Bruno Kessler (FBK) institute in Italy, and is available online [1]. The observatory uses Twitter data to quantify collective emotion, social bot pollution, and news reliability and displays this visually.

Based on the geo-tagged dataset from the US on a state and county level, (Feng and Zhou, 2020) analyzed tweets to study the daily tweeting patterns in different states. First, they could detect differences in temporal tweeting patterns and found that most state pairs have a strong linear correlation and hourly tweeting behaviors show that people tweeting more about COVID-19 during working hours. In addition, they used facial emojis to track the different types of public emotion during pandemic including an event specific subtask reporting negative emotion when the 100th and 1000th death was announced and positive when the lockdown

---

[1] https://covid19obs.fbk.eu/

measures were eased in the states.

(Lopez et al., 2020) explored the discourse around the COVID-19 pandemic and government policies being implemented. They used Twitter data from different countries in multiple languages and identify common responses to the pandemic and how these responses differ across time using text mining. Moreover, they presented insights as to how information and misinformation were transmitted via Twitter.Similarly, (Saire and Navarro, 2020) use text mining on Twitter data to show the epidemiological impact of COVID-19 on press publications in Bogota, Colombia. Intuitively, they find that the number of tweets is positively correlated with the number of infected people in the city. Most of the work discussed above focuses on analysing tweets related to single language-'English'. In our work we have designed a single model leveraging multilingual BERT for the analysis of tweets in multiple languages.

## 3 Dataset Preparation

In the last few months, a lot of work has been done focusing on collecting COVID-19 related tweets. In our work, we have trained two separate multilingual models for performing two different types of tweet classification. For training, we used publicly available open-source tweets data-

(i) For training emotion analysis model we used the 'COVID-19 Real World Worry Dataset (RWWD)'(Kleinberg et al., 2020) which is a tweet dataset consisting of 2500 long and 2500 short English COVID-19 related tweets manually annotated by human into eight classes of emotion namely, 'anger', 'disgust', 'fear', anxiety, sadness, happiness, relaxation and desire. For our experiment, we used both long and short tweets making a total of 5000 emotion labelled tweets. The dataset is prepared by researchers at University College London and collected from 6th to 7th of April 2020 from individuals living in UK.

(ii) In order to train a misinformation detection model, we have use the data publicly available at 'CoronaVirusFacts/DatosCoronaVirus Alliance Database[2]'. For our usage, we scrape 3583 false news/information belonging to four major classes namely, 'false', 'partly false', 'misleading and 'no evidence'. The data included information related to political-biased news, scientifically dubious information and conspiracy theories, mislead-

| Classes | Number |
|---------|--------|
| anxiety | 2766 |
| sadness | 714 |
| relaxation | 666 |
| fear | 468 |
| anger | 216 |

| Classes | Number |
|---------|--------|
| happiness | 78 |
| desire | 58 |
| disgust | 34 |
| total | 5000 |

Table 1: COVID-19 Real World Worry Dataset

ing news and rumors related to COVID-19.The database gathers all of the misinformation related to topics(COVID-19 cure, detection, the effect on animals, foods, travel, government policies, crime, lockdown) that have been detected by fact-checkers in more than 70 countries and includes articles published in at least 40 languages. We also used manually annotated true tweets from a public repository[3] created by the researchers (Alam et al., 2020). The dataset contained 500 tweets true and false labelled in English language. We used both for labels for training of our model.

(iii) Finally for the emotion and misinformation analysis of multilingual tweets we used the dataset published by the author in the work(Chen et al., 2020) which contains an ongoing collection of tweets IDs associated with the novel coronavirus COVID-19. Started on January 28, 2020, the current version of dataset contains 212,978,935 tweets divided into groups based on their publishing month. The dataset contains tweets in more than 40 languages containing top nine most prevalent language as- 'English','Spanish','Portuguese','Indonesian','French', 'Japanese','Thai','Hindi' and 'Italian'.

### 3.1 Dataset statistics

In this section we have provided the class distribution of each dataset. Table1 contain the distribution of eight emotion classes present in the 'COVID-19 Real World Worry Dataset(Kleinberg et al., 2020)' dataset.Table2 contain the combined final distribution of misinformation classes present in the data collected from 'CoronaVirusFacts/DatosCoronaVirus Alliance Database[4]' and here(Alam et al., 2020). Table3 shows the number of tweets from each language used for model inference and survey.

---

[2]https://www.poynter.org/covid-19-poynter-resources/

[3]https://github.com/firojalam/COVID-19-tweets-for-check-worthiness

[4]https://www.poynter.org/covid-19-poynter-resources/

| Classes | Number of tweets |
|---|---|
| false | 1272 |
| half-false | 1188 |
| no evidence | 722 |
| misleading | 519 |
| true | 382 |
| total | 4083 |

Table 2: Collected Misinformation Dataset

| Language | ISO | Number of tweets |
|---|---|---|
| English | en | 10,064 |
| Spanish | es | 10,038 |
| Portuguese | pt | 10,039 |
| Indonesian | in | 10,070 |
| French | fr | 10,097 |
| Japanese | ja | 10,006 |
| Thai | th | 10,056 |
| Hindi | hi | 10,092 |
| Turkish | tr | 10,096 |
| Total | | 90,558 |

Table 3: Language-wise Dataset Distribution

## 4 Proposed Model

In this section, we have given a detailed sequential overview of CMTA model design. We utilize the self-attention mechanism of the BERT for text feature extraction, CNN for exploiting local correlation of the data and dense layer for linear transformation.

In our proposed model, the BERT model we are adapting is a multilingual based bidirectional transformer, which is trained on 104 multiple languages. Its architecture resembles the BERT-base model with 12 encoding layers and 110M parameters and resolves the normalization issues faced in different languages. The tokenizer from MultiLingual BERT helps in tokenizing inputs of different languages by generating embeddings for the network. BERT generally gives two outputs, one pooled output also called contextual embeddings, and another hidden-states of each layer. We use both of these for further processing.

We use the dense layer or fully connected layer for linear transformation of the data by matrix-vector multiplication with Rectified Linear unit as activation, the dense layer performs a sequence of translation, rotation, and scaling based on the value of kernels and bias.

To handle the sequence data, 1D convolution

proves to be a better option. Since Conv1D can handle the spatial dimension and are known for really fast computations, they are the best efficient alternatives to traditional recurrent neural networks. Just like 2D Convolutions, we can also perform operations like padding, striding, or dilation in our architecture. In this way, Conv1D can use for hidden state values for the correlation of data.

### 4.1 Architecture and Methodology

Here we describe the phases of our architecture with their internal processes. Figure1 shows the model architecture. Generally, it is divided into four phases: tokenizing, text features extraction, linear transformation, local correlation of data, and finally, the classification after the concatenation.

#### 4.1.1 Tokenizing for Multilingual-BERT

Before everything, we need our data compatible with the network, so we will convert our textual data to numerical data using the tokenization method. Since this tokenization has to be done for our BERT model, we will be using BERT's tokenizer for multilingual data. The length of the string that should be tokenized will be limited to 512, any string greater than this much of tokens will be truncated, otherwise padded from the right. We tokenize our string to three vectors: input vectors, padding vectors, and segment vectors. These three vectors have a dimensionality of 512 each and contain the id for each token.

#### 4.1.2 Multilingual BERT: The feature Extractor

In this phase, we will be utilizing the attention mechanism of BERT on the text. Since our text is vectorized into numerical data, these vectors will be able to extract contextual features using attention mechanisms from encoder-decoder of the layers of the BERT. These values are then sent to the next encoder by a feed-forward network where Softmax is applied to normalize the output. A vector of a dimension of 768 is generated by the first encoder ,which moves through every layer of the BERT network for calculation till the last layer.

Consider $W_P$, $W_S$, and $W_I$ as weights of padding, segment and input vector respectively, according to the self-attention mechanism, if Z is an embedding vector, product of embedding vector and weights will get us the padding(P), segment(S) and input(I) matrices after training.

$$P(i) = W_P \cdot Z(i),$$

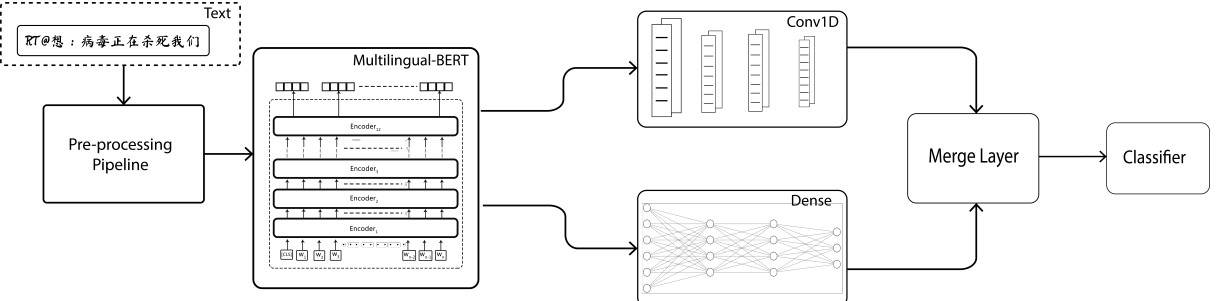

Figure 1: A detailed structure of CMTA architecture.

$$S(i) = W_S \cdot Z(i),$$

$$I(i) = W_I \cdot Z(i)$$

Matrix multiplication of segment matrices and the transpose of padding matrices will get us a new vector which will be divided by the square root of the dimension($d_k$) of embedding vector for having more stable gradients and will be passed through the softmax activation to normalize the vector, then a matrix-vector multiplication with this vector and input matrices will be applied to get the final output from the attention mechanism for next encoder layer. Followed by this operation we will get our feature extracted through the BERT. BERT will provide the contextual output along with the sequential output for further processing.

$$A(Z) = Softmax(\frac{P \times S^T}{\sqrt{d_k}}) \cdot I$$

where $A(Z)$ is output from attention layer for each input $Z$.

### 4.1.3 Linear Transformation with the dense layer

This phase comprises of Linear Transformation of data with training the architecture on pooled output also called contextual embedding. The contextual embedding from BERT is processed in this phase. The dimension of contextual embedding is 512 and is linear by nature, so a Dense layer can perform linear transformation like scaling, translation and various linear algebraic operation on the data, and with the help of the back-propagation, we normalize the value of the gradients so that this phase could be shaped perfectly to provide inferential output. To avoid vanishing gradient we use ReLU and dropout layers. Depending upon the kernel value and bias value we get our final processed output for further execution.

### 4.1.4 Convolution of hidden states

With the dense layer in action, we perform 1d convolution on the spatial output from BERT. BERT provides the hidden state f of each layer. We perform the 1-dimensional convolution on the hidden-state values along with padding them from both sides to get the same output size as the input size. We use a pair of Conv1D layers with a 1-dimensional Maximum Pooling layer of pool size 3. We use the Leaky ReLU activation function ,which enables back-propagation, even for negative input values. Consider a kernel $g$, of the length of $m$ and the input vector $f$ with length $n$, then $f \cdot g$ is a vector $p = [p_1, p_2.....p_{m+n-1}]$ where,

$$p_i = \sum_{j=1} g_j \cdot f_{i-j+1}$$

we get our resultant variable data from the convolution layers. We pass these values through the activation layer and reshape them output into a linear shape by a flattening layer.

### 4.1.5 Concatenation and the classification

In the end, we concatenate the output from the dense and flatten layer to get a single linear output. We pass this value to a pair of Dense layers for calculations of weights for the classifier. We use ReLU activation at each dense layer, and for the final layer, we apply the Softmax activation for categorical classification.

## 5 Multilingual Tweet Emotion Analysis

Public response to the COVID-19 was overwhelming. Due to the unexpected circumstances produced as a result of the sudden COVID-19 outbreak, people all around the world have a very mixed reaction. The overall effect of the whole situation was inevitable and had created a deep impact on people's psychological state. In this section, we have discussed about public emotions present in tweets and

systematically elaborated on the experimental setup of CMTA for multilingual tweet emotion analysis. Below we have described the process involving data preparation, model training ,and inference on multilingual tweet data, along with a detailed study of our findings.

## 5.1 Dataset and Proprocessing

As discussed in the section3, we used 'COVID-19 Real World Worry Dataset (RWWD)'(Kleinberg et al., 2020) for training our model. The raw tweet texts contained noises such as unnecessary symbols, misspellings ,and irregularity ,which were needed to be removed. For the preprocessing of 5000 tweets, we first created a list of Out-of-vocabulary (OOV) words which were replaced with meaningful complete words. Followed by removing URLs, blank rows, unwanted symbols, re-tweets and user-mentions. We used NLTK[5], a Python moduel for text processing removed the English stopwords and performed lemmatization of tweets. We divided the dataset into three parts: train, validation and test in the ratio of 80%/10%/10% respectively.

## 5.2 Model Setup and Training

The proposed model which we discussed in 4.1 will be trained on our dataset. We compiled the model with a Dense layer of 8 units and Softmax as the activation function as a classifier layer which will distinguish between the eight labels of emotion in a text. The model was fine-tuned by training the model on the emotion data for 10 epochs. We used Adam(Kingma and Ba, 2014) as our optimizer. For fast training, we initialized the learning rate with 1e-5 and used the method of cyclic learning rate, where the model was trained with changing learning rate value at each epoch. This helps in finding a perfect learning rate for the model training and less time to make the model learn. We used $categorical cross entropy$ loss with label smoothing(factor = 0.1) while training the model. The hyperparameter setting of the model was done using the validation data. On the test emotion dataset, our model performance score(in %) as follows: accuracy: **62.37**, precision: **65.86**, recall: **59.68** and f1: **62.58**.

## 5.3 Multilingual emotion Detection

Here we have provide the prediction of emotion class on multilingual tweets over a collection of 90,558 multilingual tweets. The bar plot shows the distribution of each eight emotion class across nine major languages.

### 5.3.1 Discussion

In the plot we can see that the the highest number of tweet are related to anxiety. This is followed by sadness and fear. We can see that majority if the tweets in all the languages are related to anxiety, sadness, relaxation and fear. Desire and disgust is the least popular class in the tweets. Anger label is more as compared to happiness. Shows us that COVID-19 have created a mixture of emotions among people but most of them are negative. In Thai we can see that the model did not find any tweets related to disgust. Spanish do not have any tweets related to happiness and relaxation.Hindi have the highest where as Thai have least number of anxiety tweets. Portuguese have the highest number of happiness tweets and English have highest relaxation tweets. Spanish and Portuguese have highest number of disgust and desire tweets respectively.

# 6 Multilingual Tweet Misinformation Analysis

The growing number of COVID-19 cases all over the world has changed the social media platform into a discussion platform where millions of people are discussing about COVID-19. This made platforms such as Twitter, Facebook as source of myths and disinformation. This is very alarming and spreading of such misinformation can cause serious public risk. In this section, we have explored misinformation present in tweets and also systematically elaborated the experimental setup of CMTA for multilingual tweet misinformation detection and analysis. Below we have described the process involving data preparation, model training ,and inference on multilingual tweet data, along with a detailed study of our findings.

## 6.1 Dataset and Proprocessing

We used publicly available misinformation and falsehood check data present at- 'CoronaVirus-Facts/DatosCoronaVirus Alliance Database[6]'. We acquired the true labeled tweets from this work(Alam et al., 2020).

---

[5]https://www.nltk.org/

[6]https://www.poynter.org/covid-19-poynter-resources/

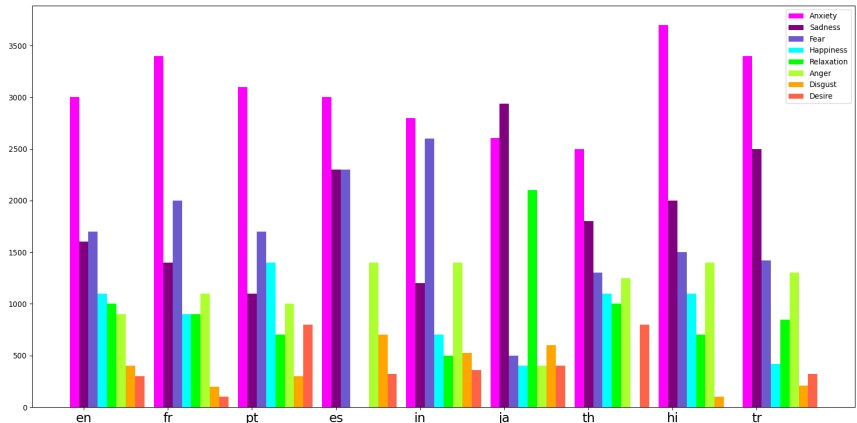

Figure 2: Distribution of labels for emotion with respect to linguistic

Most of the text data contained URLs and few unnecessary special characters which were removed. Also using NLTK[7], stopwords were removed and lemmatization of sentences were done. For tweets we did the same preprocessing as explained. For the purpose of our experiment, we distributed the dataset into three parts: train, validation and test in the ratio of 80%/10%/10% respectively.

## 6.2 Model Setup and Training

As discussed in 5.2, we did slight changes in our model's architecture since for classification of misinformation we were dependent on entities of text, so we be mainly focusing on the contextual embeddings that we received from BERT, so here we avoided the 1D convolution in the architecture of our model, and we trained the Dense layer of the network. Since there are five target variables in our training dataset, we used a dense layer of 5 units for the final classification task. The model was fine-tuned by training the model for 10 epochs. Apart from these everything remains the same, we used the cyclic learning rate method to decrease loss rate with rapid training by using the exponential decay policy. Adam optimizer was used while training, initialized with learning rate $1 \times 10^{-5}$, and the loss function utilized was $categorical crossentropy$. We feed the processed data as referred in 6.1 into our model. The hyperparameter setting of the model was done using the validation data. On the test misinformation data, our model performance score(in %) as follows: accuracy: **76.77**, precision: **84.23**, recall: **72.61** and f1: **77.96**.

---

[7]https://www.nltk.org/

## 6.3 Multilingual Misinformation Detection

Here we have provide the prediction of disinformation and truth class on multilingual tweets. The bar plot shows the distribution of each five disinformation class across nine major languages in the x-axis and there frequency on the y-axis.

### 6.3.1 Discussion

We can note that in all the nine languages the 'partly false' is in majority, followed by 'false' which have the second largest contribution in the tweets. This shows that the falsehood in the social media information is very high. Our model also predicted that the true label is very less as compared all other labels. In most of the languages the true tweets were the least. Out of misleading and no-evidence, misleading information is in tweets are more and the no-evidence is the lesser in number. It can be observed that in case of Japanese, Spanish and Indonesian the number of no-evidence class tweets are higher than the misleading tweets. Hindi have the highest where as Indonesian have the least partially-false tweets. English have the highest and Turkish have the least number of true tweets. French have the least and Indonesian have the highest number of false tweets.

## 7 Conclusion

In this paper, we presented a BERT based multilingual model for analysing COVID-19 related tweets present in more than 40 languages. We performed a detailed systematic survey for detecting emotions and disinformation spread on the social media platform- Twitter. We presented a quantified magnitude of misinformation and emotion distributed across different languages used all around the world. We strongly believe that our model can

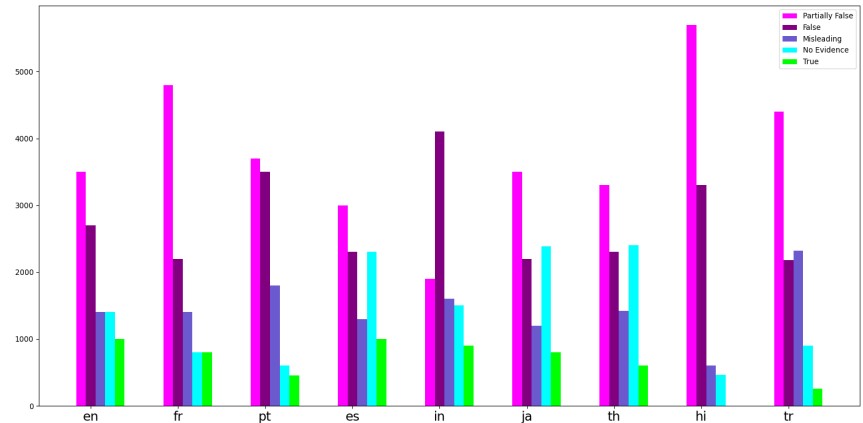

Figure 3: Distribution of labels for misinformation with respect to linguistic

help in filtration of misinformation data present in multiple languages as well as to understand public emotions and perceptions during the pandemic.

In future, we aim at collecting more annotated training data for improving our model's robustness and contextual understanding for better performance in the classification task. Task such as discovering topics and extracting keywords from multilingual tweets would be interesting.

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
