# OpenReview forum: "CMTA: A framework for Multilingual COVID-19 Tweet Analysis"
_EMNLP/2020/Workshop/NLP-COVID — Submitted to NLP-COVID19-EMNLP_

### Official Review · AnonReviewer1 · 2020-09-24
**A multilingual approach that needs to be refined**

**Rating:** 4
**Confidence:** 3

**Review:**

This work reports some experiments using the multilingual BERT model on Twitter data from different datasets in several languages: the COVID-19 Real World Worry Dataset (Kleinberg et al. 2020), the Coronavirus Facts Alliance Dataset (Alam et al. 2020), and the dataset of tweets collected by Chen et al. (2020). The supervised experiments aimed at detecting emotion (8 classes labeled in the Twitter data) and misinformation (5 classes: false, half-false, no evidence, misleading and true).

Although some of the ideas are interesting and the initiative to analyze multilingual data is compelling, several issues prevent me from accepting this work at its current state.

Authors tested the multilingual BERT under the assumption that the same pretrained model could fit all datasets, regardless of the language. As if the same multilingual model fits all. The article could have been improved if this assumption had been validated or contrasted, by comparing the results when monolingual models are used in each data set. For example, by applying the monolingual English BERT model to the English dataset, a French monolingual BERT model to the French dataset... Does a multilingual BERT model help or causes noise in some datasets?

There is a major problem with the writing style: it needs to be improved thoroughly, and several grammar, spelling and typographic errors need to be fixed.

Strengths:
- From a qualitative point of view, the analysis of Twitter data, enhanced through NLP, gives an enriching panorama of the public opinion regarding the COVID-19 disease
- The approach considers Twitter data in several languages beyond English (e.g. French, Indonesian, Japanese, Portuguese, Spanish, etc.)

Weaknesses:
- There are methodological issues regarding the experiments, which may induce to think that results are not reproducible or generalizable.
- The article needs a great deal of work to correct grammar errors, ortographic and typographic errors. Moreover, an an effort of synthesis is required: several excerpts are redundant or revolve around the same ideas about the spread of information in social media, etc.

Abstract: What does CMTS stand for? Is it a typo of "CMTA"?

Sect. 4.1.1: The authors seem to refer by "tokenization" to 2 different processing steps. The "tokenization" step refers to the segmentation of text or sentence items into individual items ("tokens"); it does not imply obtaining vectors. After tokenization, each token is then converted to a vector representation.

P. 5, sect. 4.1.3: the authors mention that they used "dropout layers", but what value exactly?

P. 6, sect. 5.1: the authors state that they "created a list of Out-of-vocabulary (OOV) words which were replaced with meaningful complete words."; I think this is not expressed correctly. What criteria did they follow to replace the OOVs? Did they use synonyms included in the vocabulary? Or do authors mean that OOV abbreviations and acronyms were expanded to full words in the vocabulary?

Regarding results, authors only reported scores on one round of experiments; the outcomes do not seem solid enough to be generalizable. A good methodological approach is initializing the model with different random seeds and test it in several experimental rounds; then, authors should report the average F-score and standard deviation.

P. 6, Sect. 5.3: "The bar plot shows..." -> Which one: Figure 2 or 3? The same happens in Sect. 6.3.

Figures 2 and 3: the colors do not distinguish well each class when printed, please use different patterns (horizontal, vertical lines...). The font size in the legend is too small.

Others (grammar, style...):

- Unify use of uppercase or lowercase letters in the title of the article.
- The acronym "CMTA" should be defined since the beginning of the article.
- Check grammar errors, e.g. (not exhaustive): "various task" -> "tasks" (p. 1); "two separate deep neural network model" -> "models" (p. 1); "have use" -> "have used" (p. 3); "contained noises" -> "noise" or "noisy content" (p. 6)
- Check ACL citation style: e.g. (p. 1): "(Matsa and Shearer, 2018)(Hitlin and Olmstead, 2018)" => "(Hitlin and Olmstead, 2018; Matsa and Shearer, 2018)"
- The authors are encouraged to split long sentences into short text fragments for the sake of clarity. For instance, the 2nd sentence of the first paragraph in the Introduction; the first sentence of Sect. 6.2...
- P. 2, Sect 1, parag. 2: Unify use of single quotes; they are sometimes used, but not in all words: ’anger’, ’disgust’, ’fear’, anxiety, sadness, happiness, relaxation ,and desire. Idem in Sect. 3, p. 3.
Regarding names of languages, no single quotes are needed (Sect. 2, p. 3): 'English' -> English, etc.
- There are many missing white spaces between words and punctuation marks (e.g. between a bracket and the next word, etc): e.g. (not exhaustive) "Twitter.Similarly" (p. 3), "topics(COVID-19" (p. 3)...
- Footnote 4 and 6 are not needed, the URL is provided in footnote 2. The same happens for footnotes 5 and 7.
- "it is divided into four phases" -> I counted 5

---

> ### Author Response · Authors · 2020-09-26
> **Fixing the paper based on the received comments**
>
> Thank you so much for your kind comments and we totally agree with everything mentioned above. We would like to answer a few doubts and clarify some points.
>
> 1. **Comparing the multilingual and monolingual BERT**: We totally agree with the reviewer concern about the multilingual BERT performance and reported results in the paper. While developing the multilingual approach, we critically brainstormed the idea of comparing the monolingual and multilingual BERT to know how the well the multilingual model is able to generalize. The first problem we faced was of reliable multilingual data unavailability. As we mentioned in the paper, we used the varified publically available dataset which was limited to English language only. So, in order to train different monolingual BERT models, we thought of translating the English dataset into 9 different languages dataset and then training separate Monolingual BERT model for each language using their respective dataset. But since the workshop deadline was very close, we dropped this idea as we were unable to complete the analysis and present it in the paper. Now, we are working and would like to add the comparative analysis in the appendix section of the paper.
>
> 2. **Paper formatting and grammatical error**: We would like to apologize for the poor paraphrasing and grammatical/formatting error present throughout the manuscript. We agree that another round of proofreading and correction is to be done. We have already taken the review's comments under consideration and fixed/added the information at the required places. The paper in the current state does not have any formatting or grammatical issue. We have also updated the bar chart for a better understanding and visualization purpose.
>
> We tried to solve both of the major comments received from two reviewers and would reflect the changes in the future version of the paper. We would again like to thank the reviews for putting their time in reviewing our manuscript and provide very insightful comments for the improvement.  At last, we believe that our work solves a very crucial task of analysing user-generated COVID-19 data in multiple languages for emotion and misinformation detection. This work takes the very common task of monolingual analysis of tweets to another level by designing a multilingual framework. In future, we aim at developing this model by collecting more dataset and hence, achieving more robust results. We would love to hear what reviews think about our actions regarding the comments.

---

### Official Review · AnonReviewer2 · 2020-09-25
**Well-motivated but lack of baseline**

**Rating:** 5
**Confidence:** 4

**Review:**

This paper proposes a multilingual framework for tweet analysis. It considers two types of tweet analysis: emotion classification and misinformation classification. The proposed framework trains a multilingual BERT model on labelled corpus and uses the trained model for emotion/misinformation detection.

The paper is well-motivated. There is no doubt that the task of emotion and misinformation analysis is important during the pandemic period, a fast-changing time period with explosion of information. Further, from a practical use point of view, the adoption of multilingual approach is potentially a good solution to address the problem of limited data availability. That is, a model trained on a corpus in one language can be transferred to use in different language context.

However, the language setting is different for emotion classification task and misinformation classification task. The emotion classification model is trained a corpus that only contains English tweets, but the misinformation classification model is trained on a multilingual dataset. This setting may be due to data availability, but I think the authors can improve the experiments by comparing the results with a monolingual BERT model. Does the multilingual BERT model make contributions?

The authors should do a thorough proof reading. The presentation of the paper is a bit difficult to follow.

---

> ### Author Response · Authors · 2020-09-26
> **Improving the paper quality based on the reviews**
>
> Thank you so much for your kind comments and we totally agree with everything mentioned above. We would like to answer a few doubts and clarify some points.
>
> 1. **Comparing the multilingual and monolingual BERT**: We totally agree with the reviewer concern about the multilingual BERT performance and reported results in the paper. While developing the multilingual approach, we critically brainstormed the idea of comparing the monolingual and multilingual BERT to know how the well the multilingual model is able to generalize. The first problem we faced was of reliable multilingual data unavailability. As we mentioned in the paper, we used the varified publically available dataset which was limited to English language only. So, in order to train different monolingual BERT models, we thought of translating the English dataset into 9 different languages dataset and then training separate Monolingual BERT model for each language using their respective dataset. But since the workshop deadline was very close, we dropped this idea as we were unable to complete the analysis and present it in the paper. Now, we are working and would like to add the comparative analysis in the appendix section of the paper.
>
> 2. **Paper formatting and grammatical error**: We would like to apologize for the poor paraphrasing and grammatical/formatting error present throughout the manuscript. We agree that another round of proofreading and correction is to be done. We have already taken the review's comments under consideration and fixed/added the information at the required places. The paper in the current state does not have any formatting or grammatical issue. We have also updated the bar chart for a better understanding and visualization purpose.
>
> We tried to solve both of the major comments received from two reviewers and would reflect the changes in the future version of the paper. We would again like to thank the reviews for putting their time in reviewing our manuscript and provide very insightful comments for the improvement. At last, we believe that our work solves a very crucial task of analysing user-generated COVID-19 data in multiple languages for emotion and misinformation detection. This work takes the very common task of monolingual analysis of tweets to another level by designing a multilingual framework. In future, we aim at developing this model by collecting more dataset and hence, achieving more robust results. We would love to hear what reviews think about our actions regarding the comments.

---

### Official Review · AnonReviewer3 · 2020-09-26
**Two BERT-based multilingual models for emotion analysis and misinformation classification on Twitter data**

**Rating:** 6
**Confidence:** 4

**Review:**

This paper proposes two separate models for multilingual emotion analysis and misinformation classification using Tweets. The authors use publicly available datasets for both tasks, and add Conv1D and Dense layers for fine-tuning the pre-trained multilingual BERT model. After briefly reporting the performance of the two models on the test sets, a bar chart is plotted for each task to show the distribution of emotion labels and misinformation classes.

Before referring to the content of the paper, I would like to strongly urge the authors to proof-read and run a grammar/format checker on the paper, as there are many formatting and grammatical issues that degrade the readability of the paper. To list a few:

1. Fix backward quotes
2. Add whitespace after the full stop
3. Check out \citep and \citet
4. Add whitespace before the left bracket and remove whitespace before the full stop.
5. Fix trivial typos that can be spotted with a spell checker.

Regarding the content of the work, I agree that multilingual misinformation detection can be useful especially in times like this when digesting wrong information could potentially impact the well-being of the person. However, due to the lack of baselines, it is difficult to evaluate the effectiveness of the proposed models. As the main contribution of the paper is the proposal of the multilingual models, the strength of the models in comparison to other baselines should have been presented clearly.

Also, it would be informative if the performance scores were given for each language as well. From my experience, the vanilla tokenizer provided by the multilingual BERT tends to over-tokenize the sentences in low-resourced languages; and their corresponding embeddings are less robust which may result in varying levels of performance for different languages. I am wondering if the distinctive patterns in the distribution of emotions for each language are due to the varying performance of the model or there is actually something distinctive about the Tweets.
If possible, a more in-depth analysis using the trained models would be nice.

---

> ### Author Response · Authors · 2020-09-27
> **Very insightful comments**
>
> We would like to thank the reviewer for putting time and reviewing our manuscript. Below we tried to answer the questions mentioned above:
>
> 1. Based on the previous comments, we have fixed the grammatical errors and typos present in the paper. It is now well-formatted and free from any errors.
>
> 2. We have also performed a comparative analysis of the mono-lingual model and multilingual model. We would add the required performance report in the appendix section of the paper.
>
> 3. The idea of adding individual performance score of languages is great. If possible, along with the existing scores, we would add the language-wise performance score within the page limit.
>
> 4. We used a random subset of a huge COVID-19 tweet corpus for the purpose of inference and generated the distribution of emotion over the multilingual tweets.  We believe that the distinctive patterns in the distribution of emotions for each language is subjective to the presence of various types of tweets in the dataset. Changing the tweets in the selected subset can affect the overall distribution of emotions.
>
> We would again like to thank the reviewer for the critical perceptive regarding the paper. We tried our best to address the comments and refine our manuscript accordingly.